# The Relationship between Binge Drinking and Binge Eating in Adolescence and Youth: A Systematic Review and Meta-Analysis

**DOI:** 10.3390/ijerph20010232

**Published:** 2022-12-23

**Authors:** Patricia Sampedro-Piquero, Clara Zancada-Menéndez, Elena Bernabéu-Brotons, Román D. Moreno-Fernández

**Affiliations:** 1Departamento de Psicología Biológica y de la Salud, Facultad de Psicología, Universidad Autónoma de Madrid, 28049 Madrid, Spain; 2Facultad de Ciencias de la Salud, Universidad Internacional de La Rioja (UNIR), 26006 Logroño, Spain; 3Facultad de Educación y Psicología, Universidad Francisco de Vitoria, 28223 Pozuelo de Alarcón, Spain

**Keywords:** adolescence, alcohol, binge drinking, binge eating, emotional eating, meta-analysis, youth

## Abstract

Adolescence and youth are critical periods in which alcohol consumption is usually initiated, especially in the form of binge drinking. In recent years, it is increasingly common to find adolescents and young people who also present binge behaviors towards unhealthy food with the aim of alleviating their anxiety (emotional eating) and/or because of impulsive personality. Despite the social and health relevance of this issue, it remains scarcely studied and more preventive research needs to be developed. Our meta-analysis study aimed to evaluate the relationship and co-occurrence of both binge behaviors during adolescence and young adulthood to clarify the link between binge drinking and eating. Selective literature search on different online databases was performed. We identified discrete but significant results regarding the direct association between binge drinking and binge eating in correlation coefficients and odds ratio. Future research should focus on the common psychological background and motives behind these problematic behaviors owing to their clinical implications for effective prevention and treatment.

## 1. Introduction

Adolescence is a particularly vulnerable period of neurodevelopment in which the incorrect management of emotional, social, and behavioral changes can lead to an unsuccessful adulthood [1]. Regarding the anatomy and function of the brain, it is well known that during this period of life the brain is still maturing, being a moment of dynamic specialization of core brain systems, particularly the frontal structures [2]. It is also a difficult period in which a high percentage of one’s decisions relies more on an emotional response or even on the social rewards rather than on a logical response. Therefore, as Hall noted in 1904, adolescence is a period of storm and stress (revised in [3]) because adolescents are characterized by a greater impulsivity, less control over impulses, behaviors, and emotions, as well as a heightened reward sensitivity [4]. As a consequence, these personality, cognitive, and behavioral patterns contribute to the emergence of risky and disruptive behaviors, such as drug abuse which is common in western countries [5].

Regarding alcohol consumption, it is a frequent problem in adolescence, and it is usually carried out on weekends involving the ingestion of large amounts of alcohol (5 or more glasses of alcoholic beverages) in short periods of time (2 h) that increase the blood alcohol concentration above 80 mg/dL [6,7,8]. Nowadays, it is estimated that 40% of people between the ages of 15 and 24 engage in this pattern of alcohol consumption, known as binge drinking (BD), which contributes to several medical complications, as well as higher risk of developing alcohol use disorder [9]. Along with the above, it has also been suggested that a significant proportion of adolescents who abuse alcohol also engage in binge eating (BE), characterized by excessive and uncontrolled consumption of unhealthy and caloric food regardless of the sensation of hunger [10]. Specifically, BE disorder was originally introduced in the appendix of the Diagnostic and Statistical Manual of Disorders 4th Edition, Text Revision [11]. It was presented as an empirically validated eating disorder which involves recurrent episodes of consuming an unambiguously large amount of food without compensatory behaviors and is associated with subjective experience of feeling a sense of loss of control. DSM-5 criteria required BE episodes at least once a week for three months which are normally preceded by an intense feeling of craving [12].

A binge-based pattern of food consumption has been termed Emotional Eating (EE), which is triggered in response to the presence or avoidance of negative mood states, and was derived from theories of obesity [13,14,15,16]. EE occurs in an effort to reduce emotional distress [17], similar to supporting evidence that individuals use alcohol to cope with negative affect [18,19]. Hence, excessive alcohol and food intake seem to share similar cue-reactivity mechanisms, with overlapping behavioral and neural activity patterns associated with problematic ingestion [20,21,22]. There is also evidence that EE can act as a predictor of BE [23,24], but it is not yet known whether BE can also act as a mediator between EE and BD. In the recent study of Escrivá-Martínez et al. (2020) [25], it was found that young students who eat more in response to negative emotions may be more likely to have higher BE scores, and engaging in BE may in turn contribute to higher alcohol consumption. Furthermore, although BE is distinguishable from BD, both problems reflect substance over-consumption and seem to share some key features such as impulsivity, neuroticism, reward dysfunction, negative emotions (e.g., anxiety, depression, guilt), and physical harm (e.g., gastrointestinal and sleep problems) [26,27,28]. In addition, both alcohol and palatable foods (high in fat, sugar, and/or salt) are calorically dense, activate brain reward circuity, and can facilitate caloric overconsumption [29,30]. Accumulating evidence has suggested that hormones like ghrelin or leptin, traditionally associated with appetite control, are also involved in alcohol-related aspects, such as craving, use, or withdrawal [31]. For instance, it was found that, in alcohol dependent subjects, concentration of leptin was significantly higher than in controls during the craving period [32]. It was also seen that levels of leptin decrease as the period of abstinence increases [33]. Regarding ghrelin, previous data suggest that this hormone signaling is related to alcohol use and may regulate alcohol seeking and consummatory behaviors [34]. Hence, it seems that neurobiological components of appetite and food craving might be similar to those of alcohol dependence, and they could constitute promising targets to develop novel treatments to manage alcohol addiction.

Owing to these similarities, in the following sections, we will revise the co-occurrence between these binge behaviors and the factors involved in their co-occurrence, and we will establish a reliable estimate of this relationship through a meta-analysis method.

### 1.1. Relationship between Binge Drinking and Binge Eating

The relationship between alcohol and food consumption is evident in comorbidity data, with substance abuse problems found in more than 30% of patients with bulimia nervosa and eating disorders among 15–56% of patients with alcohol use disorder [35]. They are often characterized as addictive because they are repetitive and uncontrollable. Data from young university students revealed that, in both men and women, the presence of alcohol and/or internet addiction as well as poor perceived health was associated with a higher prevalence of eating disorders [36]. Furthermore, this comorbidity produced more negative experiences, greater severity, greater possibility of chronicity and greater resistance to treatment of other psychiatric disorders [37]. Few studies have attempted to analyze the possible longitudinal relationship between BD and BE. One of them is the study of Measelle et al. (2006) [38], who assessed whether initial increases in depression, eating disorder, antisocial behavior, and substance abuse could predict future increases in other domains. They concluded that eating pathology predicted future substance abuse, but not vice versa. Similar results were obtained in an 8594 sample of boys and girls between 9 and 15 years of age, in which they observed that eating disorders were highly prevalent in this population and that all the types of eating disorders were associated with an early onset and frequency of BD behavior [39]. Recent findings have also suggested that, in women, both binge behaviors may be associated with difficulties of regulating emotions and facing situations involving negative emotional states [40]. In contrast, Sonneville et al. (2013) [41] did not observe that BE predicts more frequent BD in the future, while depressive symptoms and psychological distress were related to future drug use. Other studies that assessed the possible longitudinal relationship between both binge behaviors detected a unidirectional relationship between an earlier eating pathology and a later substance abuse [42]. Hence, they hypothesized that BE could promote feelings of shame and guilt, and that substance use could be a response to reduce them. Nevertheless, other studies did not observe this predictive relationship between both behaviors [39]. Therefore, results are not conclusive, and it is necessary to continue with further research to understand whether these binge behaviors are uni or bi-directionally related, as well as the directional effects of one behavior to another.

More research efforts are needed to develop valid and improved clinical measures of binge behaviors, to identify plausible transdiagnostic psychological mechanisms that would provide information on the onset, continuation, and recurrence of binge behaviors and, to recognize the consequences according to type of bingeing, possible co-occurrence of behaviors, and individuals’ sociodemographic characteristics [43,44]. Thus, the research still does not have a standardized measure for BD and BE, and future studies are needed in this field to create a comprehensive and standardized measure. Moreover, other future lines may focus on possible different paths for different populations, such as women and men. Considering all the above, our aim was to conduct a meta-analysis study to estimate, in a quantitative way, the relationship and co-occurrence of BD and BE during adolescence and young adulthood to clarify the link between both binge behaviors.

### 1.2. Psychological, Cognitive, and Social Factors Related to Binge Behaviors

Several studies have observed that both binge behaviors arise in individuals with certain psychological, cognitive, familial, and social characteristics that make them more vulnerable to developing binge episodes. Among the psychopathological aspects, both BD and BE have been associated with depressive symptomatology, anxiety, substance abuse, asocial behaviors, and suicide attempts [45,46]. These data suggest that, in young people with episodes of binge behaviors, this predisposition to act impulsively is often triggered when they experience or try to avoid negative emotions (negative urgency) [47,48,49,50]. Thus, the ability to regulate negative emotions is a highly relevant aspect at this stage of life, since it is a vital moment in which young people generally face new challenges that can be potential sources of stress (social pressure, creation of new social networks; bodily changes, responsibilities, etc.). In this regard, Kenney et al. (2013) [51] showed that college students who exhibited high levels of stress had a greater tendency to consume alcohol in large amounts and were even more likely to develop problems with alcohol consumption. Hence, stress is considered a risk factor for BE, which could even predict BE episodes in college students and become a common trigger for eating disorders [52,53]. Concerning emotion regulation, the literature notes that up to 65% of patients with eating disorders have concurrent and pro-morbid anxiety that persists even after recovery [54,55,56]. Hence, it is not surprising that negative urgency, i.e., the tendency to act rashly when distressed, is considered a predictor variable in the onset of BE among early adolescents, as well as in other risk behaviors [57,58]. A recent study has also described a relationship between recent trauma and a wide range of lifestyle habits such as BD, EE, gambling, and increased television and social media use [59]. This study suggests that trauma may enhance the risk of unhealthy behaviors in order to cope with and regulate stressful situations [60]. In terms of the biological mechanisms, it seems that cortisol is a key candidate for regulating food intake and alcohol consumption [53,60]. Other studies have also found that the probability of heavy drinking is higher than BE when there are reward and interpersonal situations related to pleasant emotions, pleasant times with others, social pressure, and conflict with others. On the other hand, both BE and heavy drinking are more associated with rather unpleasant emotions, as well as physical discomfort, urges, temptations, and testing control [61]. Specifically, when analyzing different dimensions in emotion regulation, it has been observed that adolescents with BE reported greater difficulty engaging in goal-directed behavior and controlling impulses, as well as lesser availability of emotional regulation strategies that they perceive effective compared to adolescents with restrictive eating disorders [62]. Moreover, BE seems to be related to other forms of maladaptive emotion regulation strategies, such as substance abuse and self-harm [63]. This psychological aspect has an important relevance to design possible avenues for improving treatment interventions and prevention [64,65,66].

Neuropsychological profile has been also involved in binge behaviors [67]. Thus, adolescents who perform binge behaviors show certain deficits in sustained and selective attention, inhibitory control, impulsiveness, and inflexibility to changes in new situations [68,69,70,71,72,73]. These difficulties in executive functions may contribute to both the maintenance and the development of BE and BD [70]. Even more, having difficulties in planning would lead to a clear difficulty in developing adaptive behaviors to prevent BE episodes [74]. Thus, the areas mainly affected in young people who engage in BE behaviors seem to be the so-called executive functions, or those cognitive abilities involved in planning, initiating, controlling, and regulating goal-directed behavior [75,76]. Negative emotions would reduce inhibitory control dependent on the prefrontal cortex, which seems to induce people to be more likely to eat unhealthy foods or drink uncontrollably. Emotion processing is linked to cognition since the manipulation of the one interferes with performance in the other and vice versa [77]. At the level of executive function, high-arousal information (e.g., threat) typically causes behavioral interference due to competition for resources. In this line of thought, “capacity sharing” [78] leads to executive competition for resources, a term which refers to the way executive functions such as inhibition are influenced when emotional information is encountered. Moreover, both food and alcohol act as positive reinforcers activating the mesocorticolimbic dopaminergic system which elevates positive mood and favors the maintenance of these binge behaviors [15]. Consequently, binge behavior to both substances becomes a self-reinforcing habit to attenuate negative emotions [79]. In addition, the impairment in the interpersonal relationships [80] and the lack of emotional regulation [81] often observed in these young people has been linked to the executive deficit described above [82]. For all these reasons, it is necessary to study the influence of different neuropsychological domains that can act as both predictors and variables in which to intervene during treatment. Interestingly, inhibitory functional brain networks’ abnormalities, due to neurodevelopmental and neurobiological factors, have been proposed as relevant biomarkers of adolescents at risk [83,84]. In this sense, some studies have performed neuropsychological interventions focused on inhibitory control which were associated with reductions of binge eating episodes in adolescents [85,86]. However, despite these promising results, this topic has not been yet carried out in binge drinkers, or samples of young people with both binge behaviors.

With respect to the family environment, it has been found that families which are poorly flexible, highly disengaged, and with communication problems among members served as risk factors for BE behavior [87]. Some studies have observed that young people who engage in BE behaviors perceived a reduced quality of family functioning with lower levels of cohesion, flexibility, communication, satisfaction, and a higher degree of disengagement compared to adolescents who did not show this eating pattern [87]. In addiction, and related especially to alcohol consumption, it has been found that children of parents who also have addiction problems are more likely to present excessive consumption of this drug [88]. However, there is also literature that did not find a significant association between parental alcohol consumption and different patterns of alcohol consumption in adolescence in terms of quantity and frequency of consumption [89]. Moreover, it has also been observed that a lower number of family meals, as well as critical parental comments towards weight or figure, are critical variables in increasing vulnerability to BE in young people [90]. We must also consider how momentary factors can trigger BE episodes [91]. In particular, the presence of food, as well as within-day dietary restraint, increased the risk of lack of control. Finally, it is evident that many of the potential risk family factors identified, such as a lack of family support and connectedness, may be non-specific to eating or drinking difficulties, promoting a range of negative development outcomes. Hence, insecure attachment, lower family functioning and parental involvement, parental unemployment, and parental mental disorder seem to be risk factors related to both binge behaviors during the adolescence and youth [92,93]. Interestingly, only families of adolescents who engage in both binge behaviors appeared to be highly chaotic, with problems in setting rules, a lack of clarity about roles, and decisions made impulsively rather than planned [93].

Regarding social factors, peer pressure and the influence of the social media are important variables in the onset of binge behaviors [94,95]. Thus, it has been observed that the content of social networks is poorly regulated, tending to use marketing tactics based on the vulnerability of young people to image, especially in the case of girls [95]. To date, there is little research that has investigated the profiles of youth with BE behaviors on social networks, although this could have interesting implications for treatment. A recent study found that youth with eating disorders tend to avoid posting selfies, while spending more time manipulating photos of themselves and others [96]. Following this topic of research, Nagata et al. (2020) [97] analyzed the time that more than 11,000 children aged 9 to 10 spent using different contemporary screen time modalities such as television, social media, and texting in their daily lives, and the influence this could have on the onset of BE. Their results showed that each additional hour of time using these resources was associated with 1.11 higher odds of BE disorder at 1-year follow-up because adolescents may be more prone to overeating in the absence of hunger while they are distracted in front of the screens [98,99]. Thus, although the influence of adolescents’ social networks on alcohol consumption has been studied [100,101], there are few studies that have specifically analyzed their effect on BD. In this sense, the study of Halkjelsvik et al. (2021) [102] revealed that there are some links between BD and computer gaming being both behaviors initiated by boredom. Thus, alcohol has been an outlet for thrill seeking and involves relieving boredom [103,104], and boredom has also been found to be a predictor of gambling motivation [105]. Finally, within these social factors, these binge consumption patterns are commonplace in Western society, as reflected by the elevated or increasing prevalence rates of most common binge behaviors such as binge drinking (18.2%) [44,106], binge eating (0.3% to 4.5%) [41,45], or binge watching (72%) [107], among others.

Therefore, it seems that there are many psychopathological, personality, cognitive and social factors that seem to predispose adolescents to these unhealthy and risky behaviors. It would be interesting to design prevention programs focused on them, as well as to understand the type of relationship between both binge disorders (BD and BE). This last aspect will be addressed in the following section, although more longitudinal studies would be needed.

## 2. Materials and Methods

### 2.1. Search Strategy

For this purpose, we performed a systematic review following the guidelines for systematic reviews and meta-analyses (PRISMA) [108] and the preferred reporting of the meta-analysis of observational studies in epidemiology (MOOSE) [109]. Relevant studies were identified by searching the online databases Medline and PsycINFO. The following search terms were used: (“binge eating” OR “emotional eating”) AND (“binge drinking” OR “alcohol”) AND (adolescence OR youth OR young). The selection process comprised removing duplicate articles from the results using Mendeley, as well as excluding them at abstract level when studies were not applicable to the topic of our review. To perform this abstract review, we followed and adapted the PRISMA 2020 guidelines for Abstracts checklist [110] (Figure 1). According to these guidelines, we analysed the following aspects:–Title: Identify the study as relevant for the topic of the review.–Objectives: Provide an explicit statement of the main objective(s) or question(s) the study addresses.–Methods: Specify the inclusion and exclusion criteria for the sample of the study. Specify the information about the methodology performed.–Data: Specify the methods used to analyse the results.–Results: Present results for main outcomes and if comparing groups, indicate the direction of the effect (i.e., which group is favoured).–Discussion: Provide a summary of the limitations of the results, as well as a general interpretation of the results and important implications.

### 2.2. Inclusion and Exclusion Criteria

The inclusion criteria were articles studying the relationship between BD and BE measured by standardized questionnaires in adolescents (12 to 17 years old) and the young population (18 to 25 years-old), and studies providing a correlation coefficient value or odds ratio between BD and BE, from the last 20 years (2002–2022) available up to 15 July 2022. Exclusion criteria were (a) unpublished research; (b) books, reviews, comments, editorials, master’s theses, or dissertations (Figure 1).

### 2.3. Data Extraction and Statistical Analysis

The meta-analysis was conducted using the Metafor package [111]. The correlation coefficients (Pearson’s r) extracted from studies meeting the inclusion criteria were used as the measure of effect size included in the meta-analysis. The heterogeneity of correlations across studies was assessed by the Cochran’s Q statistic and I2 statistic [112]. Potential publication bias, i.e., the tendency to publish results that were statistically significant rather than non-significant, was assessed by visual funnel plot inspection and Egger’s test [113,114].

## 3. Results

After the search process, fifteen studies were included in the meta-analysis (Figure 1). The total sample was 72,510 participants (30,932 males and 41,578 females) and the mean age 17.61 years ± 2.99 Std. Dev. Out of the ten studies analyzed for correlation index (Appendix A), seven found a significant positive correlation (*p* < 0.05) between binge drinking and binge eating (Figure 2; N = 7832; Appendix A). Given that Cochran Q Test for Heterogeneity was significant (Q9 = 97.9545, *p* < 0.0001; I 2 = 91.31%), a Random Effect (RE) Model was applied to obtain the pooled correlation coefficient (combined r = 0.12; 95% confidence intervals 0.05, 0.20; *p* < 0.005).

Each correlation coefficient is represented in a forest plot in Figure 2. Funnel plot visual inspection was carried out to assess the publication bias [113]. The symmetrical distribution of Figure 3 suggests a lack of publication bias. Plus, Egger’s regression test was not statistically significant (z = −0.3633; *p* = 0.72) and Rank Correlation Test for Funnel Plot Asymmetry showed no small-study effects (Kendall’s Tau = 0.0667, *p* = 0.86) [114,121].

We also analyzed the odds ratio between binge drinking and binge eating (Figure 4) (Appendix A). Only two of the six studies included were significant [122,123] and the effect size was rather small (OR = 1.29, 95% CI: 1.09–1.53; and 1.47, 95% CI: 1.27–1.69, respectively).

Overall, discrete results were found regarding the direct association between binge drinking and binge eating in both correlation coefficients and the odds ratio. However, the magnitude of this association could increase when adding mediation and moderation variables [25]. For instance, previous research shows that motives for drinking or eating are strongly related to binge behaviors [50,51,65,124,125,126]. Thus, rather than a direct association, future research should focus on the common psychological background and the different variables depicted in this manuscript that are behind these two binge behaviors (Figure 5).

## 4. Discussion

The objective of this study was to analyze the relationship between BD and BE in young and adolescent people. For this purpose, we performed a quantitative and systematic review to conduct a meta-analysis that could shed light on the co-occurrence between both binge behaviors in adolescents and young people. Following the established inclusion and exclusion criteria (for more details, see Results section), fifteen studies were included in the meta-analysis comprising a total sample of 72,510 adolescents and young people with a mean age of 17.61 years.

The meta-analysis displayed in this manuscript determined a rather small effect or association between these two variables (BD and BE). The positive correlation found in the RE Model (r = 0.12), even though significant, is lower than expected by previous scientific literature [10]. This result could be explained in two possible ways. On the one hand, and according to our results, these binge behaviors could be more independent of each other than has been traditionally reported in previous scientific literature. Thus, Sonneville et al. (2013) [41] also observed that BE or overeating were behavioral patterns related to starting to use marijuana and other drugs, but they were not associated with starting to binge drink frequently. Moreover, other studies found an association between alcohol consumption and personality disorders, but not between alcohol abuse and eating disorders, such as bulimia nervosa, suggesting that mediating psychological variables should be considered [127]. An alternative interpretation of the present findings is that BD and BE are different manifestations of a general vulnerability factor of emotional dysregulation, which could be shared by different behavioral and addictive disorders [27]. For instance, some authors proposed that impulsiveness acts as a general predisposition rather than a consequence of binge disorders [66,128]. In this sense, the study of Nagata et al.’s group (2000) [129] observed that multi-impulsive women engage in a variety of impulsive behaviors, of which substance misuse and binge-eating are two; a tendency towards rash-spontaneous impulsiveness appears to be a core feature that is common to all these behaviors.

Furthermore, and as we mentioned before, it is possible that other variables are mediating the relationship between these binge behaviors. Thus, several studies have pointed to different motives to initiate these sorts of unhealthy behaviors [50,51,65,124,125,126]. For instance, Luce et al. (2007) [125] found that women diagnosed with BE disorder had the highest rates of BD, compared to other eating disorders, which was related to Coping as a drinking motive. As mentioned earlier, stress can be one key factor behind both binge behaviors [51,52]. Pompili and Laghi (2017) [65] also reported similar motives for both binge behaviors, such as Reward Enhancement or Coping reasons. Interestingly, there are also positive motives, such as Social Reinforcement, which can trigger both BD and eating palatable foods [124]. Moreover, it has been observed that these motives can change when BD and BE are presented alone or together, highlighting the relevance of Coping and Social motives in the case of co-occurrence [126]. In this line of results, Escrivá-Martínez et al. (2020) [25], by a structural equation model, found that the association between BD and BE is at least partially mediated by other relevant variables, such as EE and/or external eating (predisposition to eat in response to external cues, such as sight or smell of food). Nevertheless, both maladaptive eating and alcohol use can lead to drunkorexia, defined as a condition of BD combined with self-imposed starvation similar to anorexia nervosa. It also relates to persons who use purging (as in bulimia nervosa) with the aim of decreasing caloric intake to offset the calories from alcohol [93,130]. In general, previous findings suggest that the correlation coefficients between the specific motives and both binge behaviors are higher than between the behaviors themselves [117] (Figure 2). The study of Laghi et al. (2014) [63] has also suggested a role for the ego identity in the involvement of binge behaviors. Thus, their results suggest that identity is not well defined in both binge eaters and drinkers, and they rely mostly on external social cues such as peer pressure or group demands. As identity problems have been associated with patterns of escape behavior, it is possible that binge behaviors represent a product of adolescents’ and young people’s identity process resolution. However, other authors suggest separate pathways of each binge behavior, seen as an internalized mode in eating disorder and externalized for alcohol use [131]. Finally, we cannot exclude genetic factors which could increase vulnerability. Concerning this, the study of Munn-Chernoff et al. (2013) [132] has indicated that, in women, some of the genetic risk factors that influenced vulnerability to BD also influenced vulnerability to both BE and compensatory behaviors. In any case, environmental and psychological factors might be behind the onset of binge symptoms, especially in the interaction with genetic vulnerability throughout adolescence [133,134]. This information regarding motives and moderator or mediating variables could be helpful in developing specific prevention programs and improving interventions that address risk behaviors.

According to our search strategy, our meta-analysis was composed of studies carried out with adolescents and young people of both sexes, so it is possible that this aspect could have affected the results. Thus, it is well known that eating disorders are more common in women, specifically during the youth, whereas BD is more prevalent in men as a way of entertainment and socialization [135,136]. Furthermore, it has also been observed that unhealthy alcohol consumption was higher for girls with BE episodes compared to the non-binge eating group, which was not found in the case of the men group. This last result suggests that co-occurrence between both binge behaviors could be more common in women, but not in men [136,137], so controlling gender-related differences could have enhanced the relationship between BD and BE.

Finally, we must also highlight the importance of determining the accuracy of the screening tests employed in the different studies to detect BD and BE. Thus, much of the research that has used the AUDIT, or its reduced versions, has assessed BD in very distinct manners, with most of the studies considering only the number of drinks consumed without referring the strength of the drink or without detailing the number of hours of drinking or only analyzing the consumption from the past week [138,139,140]. It is a key aspect owing to the heterogeneity of the binge-drinking group. In the case of BE, most of questionnaires have been developed to detect people with an eating disorder, while reliable measurements in an at-risk adolescent and/or young population are scarce [141].

In conclusion, this study contributes towards a deeper understanding of the relationship and co-occurrence between BD and BE in adolescents and young people. Through a review of all studies carried out to date, we observed a discrete relationship between both binge behaviors which suggest that more variables mediating BD and BE must be considered to shed light on this aspect. As a consequence, better prevention and personalized treatment approaches could be developed for vulnerable adolescents and young people who suffer from alcohol abuse and/or eating disorders.

## Figures and Tables

**Figure 1 ijerph-20-00232-f001:**
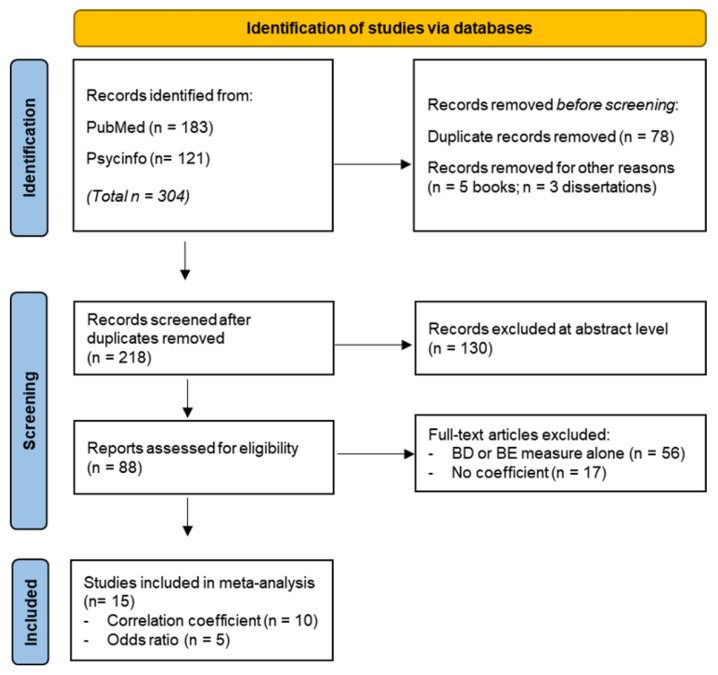
Flow diagram of the study selection process.

**Figure 2 ijerph-20-00232-f002:**
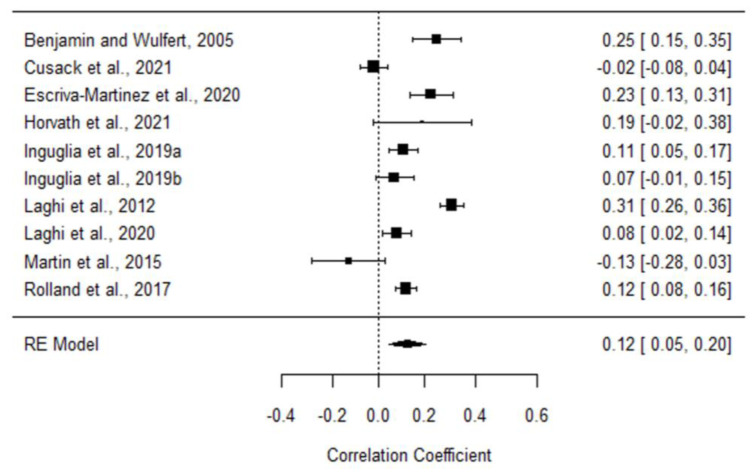
Forest plot displaying the correlation coefficients and 95% confidence intervals among the studies included in the meta-analysis. RE = Random Effects model. Escriva-Martinez et al., 2020 [25]; Rolland et al., 2017 [27]; Laghi et al., 2020 [87]; Laghi et al., 2012 [94]; Benjamin and Wulfert, 2005 [115]; Cusack et al., 2021 [116]; Horvath et al., 2021 [117]; Inguglia et al., 2019a [118]; Inguglia et al., 2019b [119]; Martin et al., 2015 [120].

**Figure 3 ijerph-20-00232-f003:**
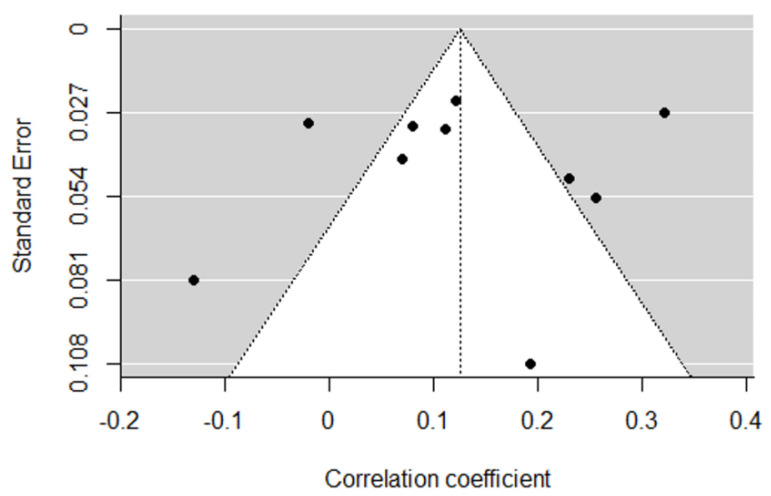
Funnel plot for publication bias analysis. Correlation coefficients are displayed in the horizontal axis and standard errors on the vertical axis. Vertical line represents the combined size effect estimated.

**Figure 4 ijerph-20-00232-f004:**
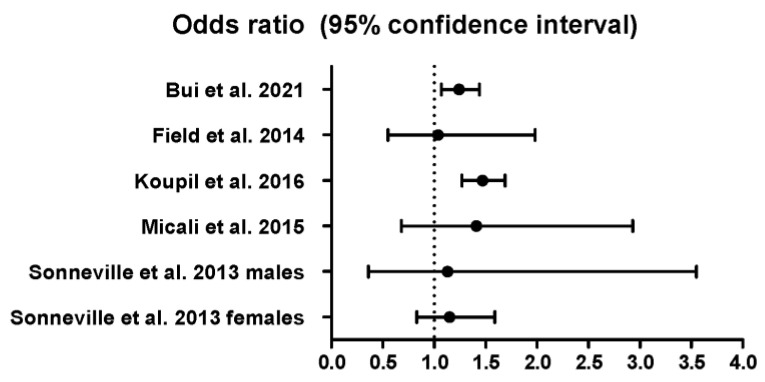
Odds ratio among the studies included in the meta-analysis. Field et al., 2014 [39]; Sonneville et al., 2013 [41]; Micali et al., 2015 [42]; Bui et al., 2021 [122]; Koupil et al., 2016 [123].

**Figure 5 ijerph-20-00232-f005:**
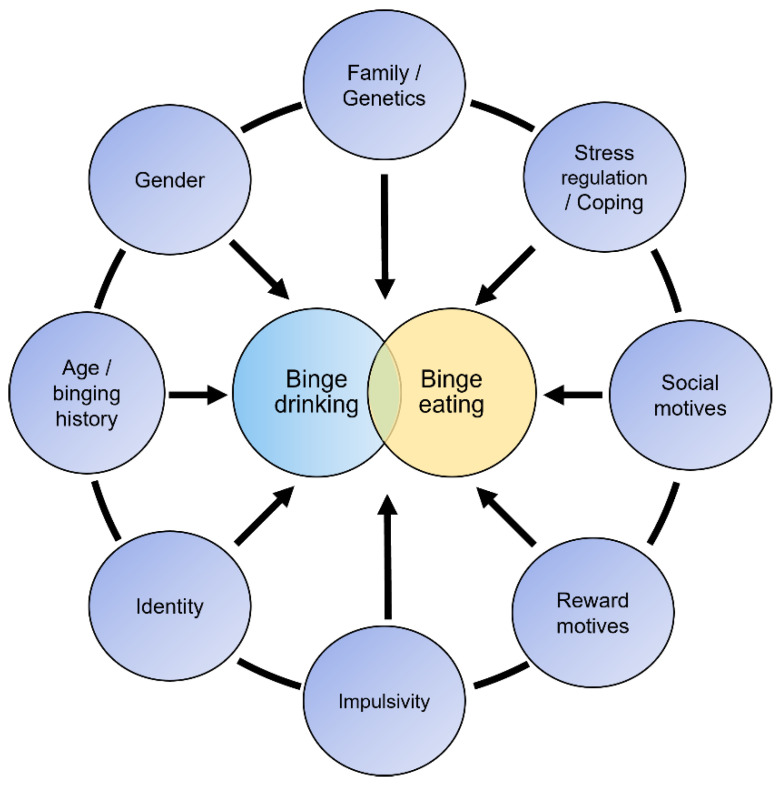
Participating variables in the relationship between BD and BE.

## Data Availability

Open Science Framework: PROSPERO (ID: 348541).

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
