# Peer review of "The Relationship between Binge Drinking and Binge Eating in Adolescence and Youth: A Systematic Review and Meta-Analysis"

_ijerph, 2022, doi:10.3390/ijerph20010232_

Round 1

Reviewer 1 Report (Previous Reviewer 2)

Thank you for making the edits! I enjoyed reading this revised manuscript and I think this study touches on an important topic.

Author Response

We would like to thank the editor and the reviewers for their valuable time and for their expert advice on how to improve our manuscript. As it has been recommended, we have included the reviewer´s suggestions in the Introduction section.

Reviewer: 1

Thank you, we are very grateful for your positive comments about our work now that all the changes have been accepted. Your suggestions were very helpful to improve our manuscript. 

Reviewer 2 Report (New Reviewer)

The manuscript entitled “The relationship between DB and BE in adolescent and youth: a systematic review and meta-analysis” affords a very interesting issue which clearly needs more investigation.

The manuscript is careful written. In my opinion the introduction section is a bit long, however since it introduces interesting and global aspect relative to the issue, I will include one interesting point which is missed. The relationships among not only hedonic but also homeostatic pathway by which satiety is regulated in both binge behaviors. For instance, there are several studies which analyze anorexigenic and orexigenic hormones such as leptin and ghrelin levels after alcohol consumption (Mehta et al. (2020); Mejía & Garciandia, 2021). This information could be included in a short way in line 59.

Finally, in order to emphasize the importance of the study,  I will start the introduction section taking into account the characteristic of adolescent from a neurophysiological point of view, since they are specially vulnerable to impulsivity, their brain is still immature, and this kind of being behaviors will lead to future health problems during adulthood.

Please review line numbers

Author Response

This manuscript is a resubmission of an earlier submission. The following is a list of the peer review reports and author responses from that submission.

Round 1

Reviewer 1 Report

The paper in its present form needs significant professional editing including the use of sub-headings. Suggestion - do not use archaic words such as "stablish" (2nd paragraph - Introduction) 

The meta-analysis, based on the last 20 years has generated 15 studies of which only 7 "found a significant positive correlation" between binge drinking and binge eating.   

The authors' considerable efforts for this paper, especially the description of the systematic review," are acknowledged.  However, the finding that a "rather small effect or association between these two variables (i.e., binge drinking and binge eating)" exists may be best presented as a concise, brief research paper.  

Reviewer 2 Report

ijerph-2040512: The relationship between binge drinking and binge eating in adolescence and youth: a systematic review and meta-analysis

This meta-analysis synthesized existing literature on binge drinking and binge eating in adolescents and young adults and quantified the relationship between these two behaviors. With the rise of alcohol/substance misuse among youth, the authors are commended for focusing on this vulnerable population. I enjoyed reading this article. Below is a list of comments that the authors may want to consider in strengthening their manuscript.

General

·      Authors used the phrase “on the other hand” multiple times throughout the manuscript. Their use of the phrase was confusing to me, as it did not seem like they were contrasting different facts/perspectives, but rather introducing additional points of view. If it is the case, authors may want to consider replacing “on the other hand” with other phrases such as besides, furthermore, also, in addition, etc.

·      It is recommended that authors use consistent language to refer to their target population. While the target population consists of both adolescents and young adults, there are times when authors only mentioned adolescents when discussing the target population as a whole, such as at the beginning of the abstract, keywords, discussion line 278, in the conclusion, etc.

Introduction

·      Authors did a very thorough review of the literature! To further improve this section, authors may want to explore ways to better the organization and make the information presented here more readily digestible to readers. As an example, after reading lines 58-61, I expected authors to first discuss the co-occurrence between binge eating and drinking, and then discuss the factors involved in the co-occurrence, but authors did the opposite. It may be helpful with the transition if the organization of the literature review mirrors the introduction of the review.

·      Another example is the organization of the first paragraph in section 1.1. Between lines 65 and 72, authors presented the idea that various psychopathologies are associated with binging, which suggests that avoidance of negative emotions may motivate binging, and thus speaks to the importance of emotion regulation to mitigate the risk of this behavior. Then authors discussed existing literature regarding the contribution of stress, anxiety, and trauma to binging, but did not expand on the role of emotion regulation. This left the paragraph feeling unfinished. Lines 77-79 touched on improving emotion regulation being a good intervention and prevention strategy, which could be a good conclusion, but this sentence was in the middle of the paragraph and seemed to be out of place with the current organization. Also, as a side note, authors first introduced the concept of negative urgency in line 69 but did not provide a formal definition of this term until line 83. I recommend authors revise this paragraph.

·      Can authors provide more explanations of the statement “negative emotions would reduce inhibitory control” (lines 104-105)?

·      It is unclear how the family environment relates to the co-occurrence of binge drinking and eating. According to authors, family environment and family meals influence binging eating, but parental alcohol use influences binge drinking. More clarification is needed, especially about shared familial factors that influence both binge drinking and eating.

·      In line 172, authors referred to college students as boys and girls, and in line 181, referred to 9- to 15-year-olds as women. Authors may want to use boys/girls to refers to adolescents and men/women to refer to adults.

Method

·      In section 2.2, it would be helpful if authors could specify the age range for “adolescents and young population.”

·      Is there a particular reason why the search was limited to the last 20 years?

Results

·      I would love to know more about participant characteristics, if available, such as gender, standard deviation of participant age, SES, etc.

Discussion

·      More clarifications are needed regarding the roles of ego identity and genetic risk factors. Based on authors’ description, ego identity and genetic risk factors seem to be more of moderating variables than mediating variables. In addition, if genetic factors can influence both binge drinking and binge eating, does it mean that the correlation between these two behaviors should be high?

·      I wonder whether authors are referring to their own discoveries or citing other papers for the statement they made in lines 338-340.

Reviewer 3 Report

In the paragraph 2. Materials and Methods, 2.1 Search Strategy,

it is suggested to develop and justify in the text the selection process as shortly described in the Figure 1. For example, what criteria have been used for excluding Records at abstract levels.

2.2. Inclusion and exclusion criteria

Row 226: The exclusion criteria ‘non-English or Spanish language publications’ is not an objective criterion. We can understand the choice of such criteria, but it means that some important studies are excluded…

Paragraph 3. Results

Row 236: ‘’After the search process, fifteen studies were included in the meta-analysis (Figure 2)’’, but the figure 2 concerned only the 10 studies for correlation index.!

Row 248: ‘Egger's regression test was not statistically significant’.  It should be interesting to consider alternative tests as Begg’s test, Deeks’ test or at least justify the exclusive use of Egger’s test.

4. Discussion

Row 273: 4. ‘’Authors The objective of this study was to provide’’ what means here Authors?